# Rokubacteria in Northern Peatlands: Habitat Preferences and Diversity Patterns

**DOI:** 10.3390/microorganisms10010011

**Published:** 2021-12-22

**Authors:** Anastasia A. Ivanova, Igor Y. Oshkin, Olga V. Danilova, Dmitriy A. Philippov, Nikolai V. Ravin, Svetlana N. Dedysh

**Affiliations:** 1Winogradsky Institute of Microbiology, Research Center of Biotechnology of the Russian Academy of Sciences, 119071 Moscow, Russia; ivanovastasja@gmail.com (A.A.I.); ig.owkin@gmail.com (I.Y.O.); olga.v.danilova10@gmail.com (O.V.D.); 2Papanin Institute for Biology of Inland Waters, Russian Academy of Sciences, 152742 Borok, Russia; philippov_d@mail.ru; 3Institute of Bioengineering, Research Center of Biotechnology of the Russian Academy of Sciences, 119071 Moscow, Russia; nravin@biengi.ac.ru

**Keywords:** Rokubacteria, boreal wetlands, eutrophic fens, bacterial diversity, unknown metabolic potential, incubation with methane

## Abstract

Rokubacteria is a phylogenetic clade of as-yet-uncultivated prokaryotes, which are detected in diverse terrestrial habitats and are commonly addressed as members of the rare biosphere. This clade was originally described as a candidate phylum; however, based on the results of comparative genome analysis, was later defined as the order-level lineage, *Rokubacteriales*, within the phylum *Methylomirabilota*. The physiology and lifestyles of these bacteria are poorly understood. A dataset of 16S rRNA gene reads retrieved from four boreal raised bogs and six eutrophic fens was examined for the presence of the *Rokubacteriales*; the latter were detected exclusively in fens. Their relative abundance varied between 0.2 and 4% of all bacteria and was positively correlated with pH, total nitrogen content, and availability of Ca and Mg. To test an earlier published hypothesis regarding the presence of methanotrophic capabilities in Rokubacteria, peat samples were incubated with 10% methane for four weeks. No response to methane availability was detected for the *Rokubacteriales*, while clear a increase in relative abundance was observed for the conventional *Methylococcales* methanotrophs. The search for methane monooxygenase encoding genes in 60 currently available *Rokubacteriales* metagenomes yielded negative results, although copper-containing monooxygenases were encoded by some members of this order. This study suggests that peat-inhabiting *Rokubacteriales* are neutrophilic non-methanotrophic bacteria that colonize nitrogen-rich wetlands.

## 1. Introduction

The Rokubacteria is one of the deeply branching lineages of prokaryotes, which are often addressed as members of the rare biosphere [1,2,3]. The existence of this bacterial group was first recognized by Lipson and Schmidt [4], who retrieved the corresponding 16S rRNA gene sequences (GenBank accession numbers AY192281 and AY192282) from an alpine meadow of the Colorado Rocky Mountains. These sequences clustered together with two other environmental 16S rRNA gene sequences (AF234132 and AF428647), which were earlier obtained from an Australian arid soil and a lake sediment in China, and formed a deeply rooting, phylum-level bacterial lineage, which was originally referred to as the Candidate phylum SPAM (for Spring Alpine Meadow) [4]. The name *Candidatus* Rokubacteria was proposed in 2016 for the phylogenetic lineage defined by the metagenome bin CSP1-6, which was assembled from the sediment sample of the Rifle DOE Scientific Focus Area in Colorado [2]. This lineage was equally divergent (82% 16S rRNA gene sequence similarity) from the *Deltaproteobacteria*, *Firmicutes*, *Nitrospirae* and the *Candidate* phylum NC10. The fact that *Candidatus* Rokubacteria represents the same phylogenetic lineage as SPAM was noticed by Becraft et al. [5]. Notably, in treeing analyses that imply normalization of evolutionary distances, *Candidatus* Rokubacteria formed a common group with the candidate phylum NC10 [6,7]. Therefore, based on the results of the comparative genome analysis, this bacterial group was recently defined as the order-level lineage, *Rokubacteriales*, within the phylum *Methylomirabilota,* the class *Methylomirabilia* [7]. The second order within this class is the *Methylomirabilales* (also referred as NC10), which accommodates bacteria that couple anaerobic oxidation of methane with the reduction of nitrite to dinitrogen [8].

Rokubacteria inhabit a wide variety of diverse terrestrial habitats. Thus, *Rokubacteriales*-related 16S rRNA gene sequences were detected in crop soils [9,10,11], copper mine soil [12], the subsurface oxic sediments [13], Brazilian Amazon rainforest soil [3], virgin soils of southern Vietnam tropical forests [14] and Korean pine forests [15], as well as in wetlands [16,17].

Despite the wide distribution of Rokubacteria, all attempts to isolate these microorganisms to date remain unsuccessful. All insights into the characteristic features of these enigmatic organisms were made by the cultivation-independent molecular approaches, including single-cell genomics and metagenomics [3,5]. These analyses showed that rokubacterial genomes are large (6–8 Mb) and have high GC content. At the same time, the cell sizes of these bacteria assessed using the fluorescence-activated cell sorting are small, 0.3–0.4 µm in diameter [5]. The presence of large genomes in small cells may imply extensive DNA packaging or dormancy. Notably, Rokubacteria are characterized by an intriguing high genomic heterogeneity among individuals, with no environments identified to date with near-clonal populations. This feature may present certain challenges to future studies [5].

Several recent studies have illuminated the possible metabolic and ecological properties of Rokubacteria [3,16,18]. Anantharaman and co-authors discovered that these bacteria contain some of the earliest evolved dissimilatory sulfite reductases (DsrAB) [18]. Rokubacterial DsrAB-encoding genes represent a novel deep-branching lineage on the tree among the sulfate/sulfite-reducing microorganisms. Besides *dsrAB* genes, Rokubacteria also possess *apr*, *sat*, and *qmo* genes that are required for sulfate reduction or sulfite oxidation [18]. In addition, a number of novel putative isopropanol dehydrogenases affiliated with the Rokubacteria were identified while studying wetland sediments [16]. Therefore, it cannot be ruled out that Rokubacteria may utilize ethanol and/or isopropanol as electron donors, which suggest their potential contribution to alcohol cycling in wetlands [16]. The most intriguing hypothesis regarding the metabolic capabilities of the Rokubacteria, however, was published by Kroeger and coauthors, who assembled and examined a rokubacterial metagenome-assembled genome (MAG), named Amazon-R-15-13, from Brazilian Amazon rainforest soil [3]. One of the unique features of Amazon R-15-13 was the presence of a gene operon with a structural similarity to that of particulate methane monooxygenase (pMMO), a key enzyme of methanotrophic metabolism. The corresponding enzyme from Amazon R-15-13, however, clustered together with monooxygenases from two non-methanotrophic actinobacteria, *Nocardioides luteus* and *Streptomyces theroautotrophicus*, and revealed only a very distant relationship to true pMMO from characterized methanotrophic bacteria.

This study was initiated in order to examine the diversity patterns of the *Rokubacteriales* in boreal wetlands and to verify the hypothesis regarding the presence of methanotrophic capabilities in these bacteria in incubation experiments. All currently available metagenomes of the *Rokubacteriales* were also examined for the presence of pMMO-encoding genes as well as the genes encoding evolutionary related copper-containing monooxygenases.

## 2. Materials and Methods

### 2.1. Analysis of Microbial Diversity in Boreal Wetlands

The comparison of microbial diversity patterns in two types of boreal wetlands, acidic razed bogs and neutral eutrophic fens, was performed using the 16S rRNA gene sequence dataset retrieved by Dedysh et al. (2021) and deposited in Sequence Read Archive (SRA) under the accession numbers SRR11280489-SRR11280524 (Bioproject PRJNA610704). The set of examined wetlands included four raised bogs (Shichengskoe, Piyavochnoe, Barskoe and Alekseevskoe) and six eutrophic fens (Shichengskoe, Piyavochnoe, Rodionskoe, Ileksa, Povreka and Charozerskoe) located in the Vologda region of European North Russia, within the zone of middle taiga. Detailed descriptions of the sampling sites and the sampling procedure are reported elsewhere [17,19]; characteristics of the examined peatlands are also provided in Table 1.

### 2.2. Peat Sampling for Incubation Studies

To examine the response of peat-inhabiting representatives of the *Rokubacteriales* to methane availability, one additional batch of peat samples was collected in June 2021 from the eutrophic fen site of the mire Shichengskoe (59°56′56″ N, 41°16′59″ E). The samples were transported to the laboratory, homogenized and used for the determination of methane oxidation activity, incubation experiments and molecular analyses.

### 2.3. Methane Incubation Experiments

Weighted portions of peat (10 g, water content 91%) sampled from the fen Shichengskoe were placed in 160 mL glass flasks, which were then sealed hermetically. Methane (CH_4_) was injected in the flasks up to the concentration of ~1000 ppm and the flasks were incubated at room temperature for 24 h. Samples of the gas phase (0.5 mL) were taken from the flasks periodically and analyzed for CH_4_ concentration on a Kristall 5000 chromatograph (Khromatek, Yoshkar-Ola, Russia). The experiments were made in triplicate. The rates of methane oxidation by the samples were calculated in mg CH_4_ g^−1^ of wet peat h^−1^. After these activity measurements, methane was added to the flasks up to the concentration of 10% (*v/v*) in the headspace. The flasks were then incubated at room temperature for 4 weeks. At the end of each week, the flasks were flushed with ambient air using a sterile filter (0.22 µm) to remove remaining CH_4_ and accumulated CO_2_, and methane was re-injected up to the concentration of 10% (*v/v*), in order to keep high methane availability during the whole incubation period. Portions of peat from each incubation flask were taken before and after incubation with CH_4_ and used for the molecular analysis.

### 2.4. DNA Extraction and 16S rRNA Gene Sequencing

Total genomic DNA was isolated from peat taken before and after incubation in flasks with CH_4_ using a Power Soil DNA isolation kit (MO BIO Laboratories, Inc., Carlsbad, CA, USA). PCR amplification of 16S rRNA gene fragments comprising the V3–V4 variable regions was performed using the universal primers 341F (5′-CCTAYG GGDBGCWSCAG) and 806R (5′-GGA CTA CNVGGG THTCTAAT) [20]. The obtained PCR fragments were bar-coded and sequenced on Illumina MiSeq (2 × 300 nt reads). Pairwise overlapping reads were merged using FLASH v.1.2.11 [21].

### 2.5. Bioinformatic Analyses of Microbial Community Structure

Both the datasets of 16S rRNA gene sequences from Dedysh et al., 2020 and from the current study were analyzed with *QIIME 2* v.2019.10 (https://qiime2.org) [22]. The DADA2 plugin was used for sequence quality control, denoising and chimera filtering [23]. Operational taxonomic units (OTUs) were clustered applying the VSEARCH plugin [24] with open-reference function using the Silva v. 132 database [25,26] with 97% identity. Taxonomy assignment was performed using BLAST against the Silva v. 132 database with 80% identity.

### 2.6. Correlations between Microbial Groups and Chemical Properties of Peat

Graph Pad Prism v. 6.0.1 (GraphPad Software, San Diego, CA, USA, www.graphpad.com) package was used to calculate Pearson correlation coefficients between abundances of the taxonomic groups and chemical properties of peat samples. Tests were considered significant if they had a *p*-value < 0.05.

### 2.7. Analyses of the Publicly Available Rokubacterial Metagenomes

All available rokubacterial MAGs were retrieved from the GenBank using taxid and combined into the single database. The entire pool of OTUs identified in this study was compared against this Rokubacteria database using BLAST (v. 2.9.0) [27] (accessed on 15 November 2021) with a sequence identity threshold of 97%. The constructed database was annotated using Prokka [28]. Amino acid sequence of the putative monooxygenase from the assembly Amazon-R-15-13 was taken as a query for tblastn searches against the Rokubacteria database with a sequence identity threshold of 30%. All valid blast hits were included into the phylogenetic analysis. A genome-based tree for members of the Rokubacteria was reconstructed using the Genome Taxonomy Database and GTDB-toolkit (https://github.com/Ecogenomics/GtdbTk (accessed on 16 November 2021)) [29], release 04-RS89. The maximum likelihood phylogenetic tree was constructed using MegaX software [30]. Functional annotation of CDSs in MAGs was performed on the basis of the results of BLASTP searches [27] against the NCBI (National Center for Biotechnology Information) nonredundant database and the KEGG (Kyoto Encyclopedia of Genes and Genomes) database [31].

## 3. Results and Discussion

### 3.1. Microbial Community Composition in Boreal Wetlands

The pool of 16S rRNA gene reads examined in this study included 1,024,783 sequences (average length, ~440 bp), which were retrieved from the peat samples and retained after quality filtration, denoising and removing chimeras. The microbial community composition in raised bogs was clearly different from that in fens (Figure 1). Thus, archaeal populations in raised bogs were represented by members of the *Euryarchaeota* and *Taumarchaeota*, while fens were colonized by the *Euryarchaeota*, *Crenarchaeota* and *Nanoarchaeaeota*. The patterns of bacterial diversity in raised bogs and fens were also different. Raised bogs were dominated by the *Acidobacteria* (from 30.0 ± 0.4% to 42.4 ± 2.9% of total reads retrieved from the Alekseevskoe and Barskoe fens, respectively, mean ± SE). The next numerically abundant bacterial groups in bogs were the *Planctomycetes* (11.0 ± 2.7% and 19.2 ± 0.4% in the Piyavochnoe and Barskoe), *Proteobacteria* (10.7 ± 2.0% and 15.7 ± 2.8% in Alekseevskoe and Piyavochnoe) and *Verrucomicrobia* (11.3 ± 1.2% and 14.8 ± 0.5% in Barskoe and Piyavochnoe). By contrast, the bacterial communities in fens were dominated by the *Proteobacteria* (from 21.1 ± 3.2% to 30.6 ± 2.1% of total reads retrieved from the Piyavochnoe and Rodionskoe fens, respectively) and *Chloroflexi* (from 15.7 ± 1.8% to 30.2 ± 3.7% of total reads retrieved from the Shichengskoe and Ileksa fens, respectively). Other major groups were the *Acidobacteria* (7.2 ± 0.1% to 22.3 ± 1.2% retrieved from the Ileksa and Shichengskoe) and *Patescibacteria* (6.0 ± 0.5% to 18.8 ± 2.3% in Povreka and Piyavochnoe).

Notably, members of the *Methylomirabilota* were detected in all studied fens but were absent from raised bogs (Figure 1). These bacteria were especially abundant in Charozerskoe and Shichengskoe fens (4.2 ± 0.3% and 3.3 ± 0.3% of total reads) but represented only a minor group in fens Povreka and Ileksa (0.3 ± 0.1% of total reads in both wetlands). The relative abundances of Rokubacteria determined in our study are in agreement with previous reports. Bacteria of this group were low in abundance at almost every site where they were identified [11,32], and often were represented by a single 16S rRNA gene sequence. The two so far reported exceptions were the microbial community in a grass root zone in the Angelo Coast Range Reserve, California, where Rokubacteria constituted ~10% of all prokaryotes [33] and the microbial community in virgin soils of southern Vietnam tropical forests, where the relative abundance of these bacteria reached 6% in the lower soil horizons [14].

### 3.2. Diversity and Most Abundant OTUs of Methylomirabilota

Since Rokubacteria were only detected in eutrophic fens, our further analyses were focused on the wetlands Shichengskoe, Piyavochnoe, Rodionskoe, Ileksa, Povreka and Charozerskoe. The pool of *Methylomirabilota*-affiliated 16S rRNA gene fragments obtained in our study included 7837 reads. The highest numbers of reads were received from Shichengskoe and Charozerskoe fens (3458 and 2647, respectively) and the lowest from Ileksa and Povreka (247 and 244). Taxonomic analysis revealed that peat samples from Ileksa and Povreka fens were mostly represented by uncultured bacteria from the family *Methylomirabilaceae* of the order *Methylomirabilales* (98% and 100% of all *Methylomirabilota*-affiliated reads). On the contrary, samples from Shichengskoe and Charozerskoe fens were dominated by uncultivated members of the order *Rokubacteriales* (96% and 93%, respectively). In peat samples from the fens Piyavochnoe and Rodionskoe, the proportions of *Rokubacteriales*-/*Methylomirabilales*-like reads were as follows: 59/41% and 55/45%, respectively (upper panel in Figure 1).

In total, 137 OTUs of *Methylomirabilota* determined at 97% sequence similarity were identified in our study. Of these, 1 OTU was present in all fens, 3 OTUs were detected in four samples, 3 OTUs were detected in three samples, 8 OTUs were present in two samples and the remaining 122 OTUs were specific for each fen (Figure 2). The list of the most abundant OTUs comprising ≥ 0.1% of all reads retrieved from the corresponding fen is given in Table 2 and is shown in the phylogenetic tree in Figure 3. The most abundant OTUs were represented by uncultivated members of the *Rokubacteriales* (OTUs No 1, 3). However, OTU 8, which was present in all studied fens, belonged to the family *Methylomirabilaceae*. Overall, among the fourteen most represented OTUs, three OTUs belonged to the family *Methylomirabilaceae* and eleven OTUs affiliated with the order *Rokubacteriales* (Table 2, Figure 3).

All rokubacterial MAGs available in the GenBank were combined into the single database, which also contained MAGs of higher quality from GTDB. OTUs retrieved from the examined fens were compared against this database using BLAST with a species-level sequence identity threshold of 97%. Nine blast hits against GTDB metagenomes affiliated with MAGs assembled from grassland soil and aquifer well ecosystems (Figure 4). The remaining blast hits were represented by the GenBank MAGs from meadow soil, wetland sediment, freshwater lake and hot spring sediment (not shown).

### 3.3. Correlations between Chemical Properties of Peat and the Abundance of Methylomirabilota

To reveal the influence of several chemical characteristics of peat on the relative abundances of *Methylomirabilota* groups, Pearson correlation analysis with further significance test was performed. Relative abundance of *Rokubacteriales* correlated positively with pH, total nitrogen (TN), Ca and Mg availability, and negatively correlated with total organic carbon (TOC), sulfate, Fe and P availability (Figure 5). Notably, a strong positive correlation with pH, TN, Mg and Ca availability, as well as a negative correlation with TOC, was observed for the representatives of *Methylomirabilaceae* (Figure 5).

### 3.4. Microbial Community Response to Incubation with Methane

To examine the response of peat-inhabiting *Rokubacteriales* to methane availability, peat samples from the fen Shichengskoe were incubated with 10% (*v/v*) CH_4_ for 4 weeks. Incubation with methane resulted in a pronounced increase of methane-oxidizing activity of peat samples from 0.20 ± 0.02 to 0.40 ± 0.09 mg CH_4_ g^−1^ h^−1^. The bacterial community composition in these samples was analyzed before and after incubation with methane. The original methanotroph community was composed of representatives of the order *Methylococcales*, particularly uncultivated *Crenothrix* bacteria, and members of the family *Beijerinckiaceae* (Figure 6). The strongest positive response to methane availability was detected for members of the order *Methylococcales*, which were represented mostly by *Methylovulum* spp., *Methylomonas* spp., *Methyloglobulus* spp. and *Methylomagnum* spp. (Figure 6). In contrast, the relative abundances of the *Beijerinckiaceae* and *Crenothrix* declined after incubation with methane, which was also true for members of the *Methylomirabilaceae*. No statistically significant response of the *Rokubacteriales* to CH_4_ availability was observed in this incubation experiment.

### 3.5. Search for the Presence of Genomic Determinants of Methanotrophy in Rokubacterial MAGs

To verify the hypothesis regarding the occurrence of methanotrophic capabilities in Rokubacteria, we inspected all metagenomes of these microorganisms available in the GenBank for the presence of genes encoding pMMO, a key enzyme of aerobic methanotrophy. The latter is encoded by the *pmoCAB* gene cluster; the *pmoA* is used as a functional gene marker for detecting aerobic methanotrophs in the environment [34,35]. The pMMOs belong to the superfamily of Cu-containing membrane-bound monooxygenases (CuMMOs), which also includes ammonia monooxygenases (AMO) and a number of short-chain alkane and alkene monooxygenases [36,37]. The genes coding for CuMMOs are usually organized in the same way as pMMO (*xmoCAB* gene cluster) [37]. It should be taken into account, however, that the search for the presence of *xmoCAB* operons in rokubacterial MAGs has some limitations due to the lack of complete rokubacterial genomes.

The amino acid sequence (WP_008359136.1) deposited in GenBank as a putative MMO from the MAG Amazon-R-15-13 [3] was used as a query against the database of rokubacterial metagenomes. Only one blast hit displayed a reliable identity threshold (89%). This amino acid sequence belonged to the rokubacterial MAG AR15 retrieved from a meadow soil (GCA_003220675) (Figure 7). Both XmoA sequences were organized in clusters with XmoB and XmoC. The evolutionary relationships of rokubacterial XmoA sequences among other members of the CuMMO superfamily were determined by means of phylogenetic analysis (Figure 7). Notably, XmoA sequences from rokubacterial MAGs did not cluster with PmoA sequences of true methanotrophs from the *Alphaproteobacteria*, *Gammaproteobacteria*, *Verrucomicrobia* and candidate phylum NC10 [8,38,39]. Instead, two rokubacterial XmoA sequences clustered together with XmoA sequences from several members of the *Actinobacteria,* which are known for the ability to utilize short alkanes but not methane. Three of these actinobacteria, namely *Nocardioides* sp. strain CF8 and *Mycolicibacteria rhodesiae* strains NBB3 and NBB4, possess putative CuMMOs (pBMOs), which function as butane monooxygenases [40,41]. The fourth actinobacterium in this clade, *Rhodococcus* sp. ZPP, is capable of growth on propane, which is dependent on the presence of CuMMO genes [42]. Thus, clustering of both rokubacterial XmoA sequences with those from *Actinobacteria* suggests that Rokubacteria may potentially act as short chain alkane oxidizers.

Although CuMMOs display clear specialization for one particular substrate, they can co-oxidize several substrates, including methane and alkanes containing up to five carbons [43]. We, therefore, looked also for the occurrence of C1-metabolism genes in two selected rokubacterial MAGs, AR15 and Amazon-R-15-13. Rokubacteria bacterium AR15 possesses PQQ-dependent methanol dehydrogenase (MDH), which displays 57% identity to that in *Methylococcus capsulatus* Bath. MDH is most likely absent from Amazon-R-15-13, since the best blast hit revealed only 27% identity to MDH sequence in AR15. The genes coding for formate dehydrogenase are present only in AR15. Formaldehyde assimilation may potentially function via the serine cycle. Completeness of the latter is difficult to confirm due to the incompleteness of both MAGs. However, phosphoenolpyruvate carboxylase, a functionally important enzyme, is absent from both AR15 and Amazon-R-15-13.

On the other hand, utilization of short chain hydrocarbons is supported by the presence of genes encoding alcohol and aldehyde dehydrogenases as well as genes coding for the pathway of β-oxidation of fatty acids. The fatty acids produced by aldehyde oxidation are further metabolized by β-oxidation, generating Acyl-CoA, which enters the tricarboxylic acid cycle [44]. Therefore, AR15 and Amazon-R-15-13 most likely represent bacteria that utilize short chain hydrocarbons.

## 4. Conclusions

In summary, our study identified neutral eutrophic fens as one of the common habitats for bacteria of the order *Rokubacteriales*. Relative abundance of these microorganisms correlated positively with pH, total nitrogen, Ca and Mg availability. No response to methane availability was detected for the *Rokubacteriales*, while a clear increase in relative abundance was observed for the conventional *Methylococcales* methanotrophs. The search for methane monooxygenase encoding genes in the publicly available *Rokubacteriales* metagenomes yielded negative results. Some members of this bacterial order may possess Cu-containing membrane-bound monooxygenases, but the exact substrate specialization of these enzymes in *Rokubacteriales* remains unclear and requires experimental verification. Apparently, peat-inhabiting *Rokubacteriales* are non-methanotrophic bacteria that colonize nitrogen-rich wetlands.

## Figures and Tables

**Figure 1 microorganisms-10-00011-f001:**
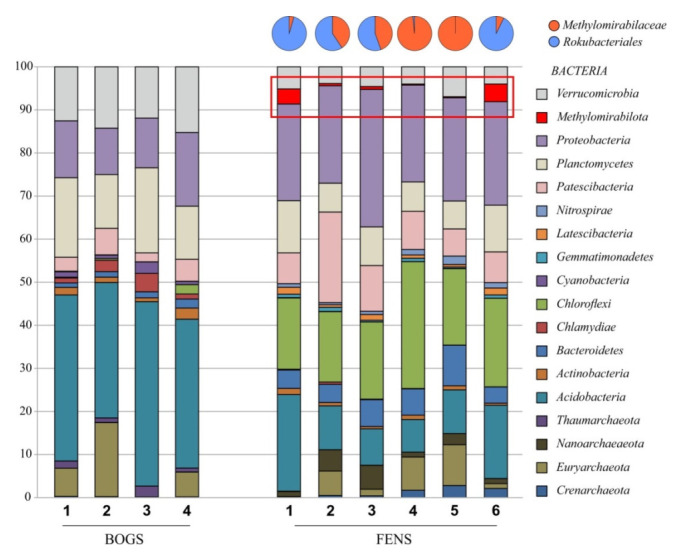
Bacteria community composition in bogs (1-Shichengskoe, 2-Alekseevskoe, 3-Barskoe, 4-Piyavochnoe) and fens (1-Shichengskoe, 2-Piyavochnoe, 3-Rodionskoe, 4-Ileksa, 5-Povreka, 6-Charozerskoe). The composition is displayed at the phylum level. The relative abundance values represent averages of three replicate data sets. Phyla with the relative abundance of at least >0.1% for one ecosystem are shown.

**Figure 2 microorganisms-10-00011-f002:**
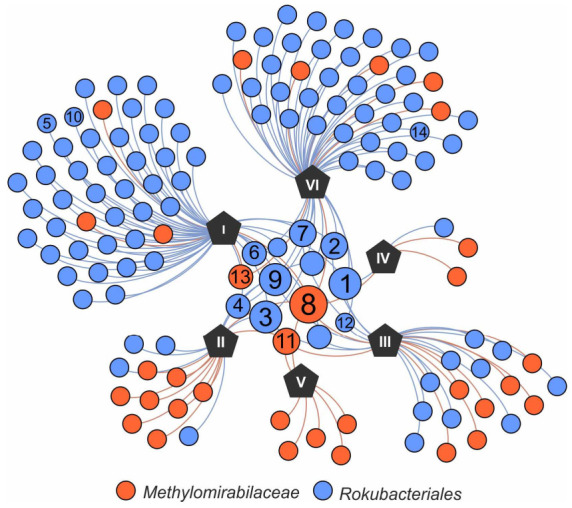
Network graph of six fen sites (black pentagons, I-Shichengskoe, II-Piyavochnoe, III-Rodionskoe, IV-Ileksa, V-Povreka, VI-Charozerskoe) based on the presence of rokubacterial operational taxonomic units (OTUs). The size of the OTU nodes is weighted according to the number of fens in which the particular OTU occurred. The OTU numbers in circles match those in Table 2. The OTUs affiliated with *Methylomirabilaceae* and *Rokubacteriales* are indicated by red and blue, respectively.

**Figure 3 microorganisms-10-00011-f003:**
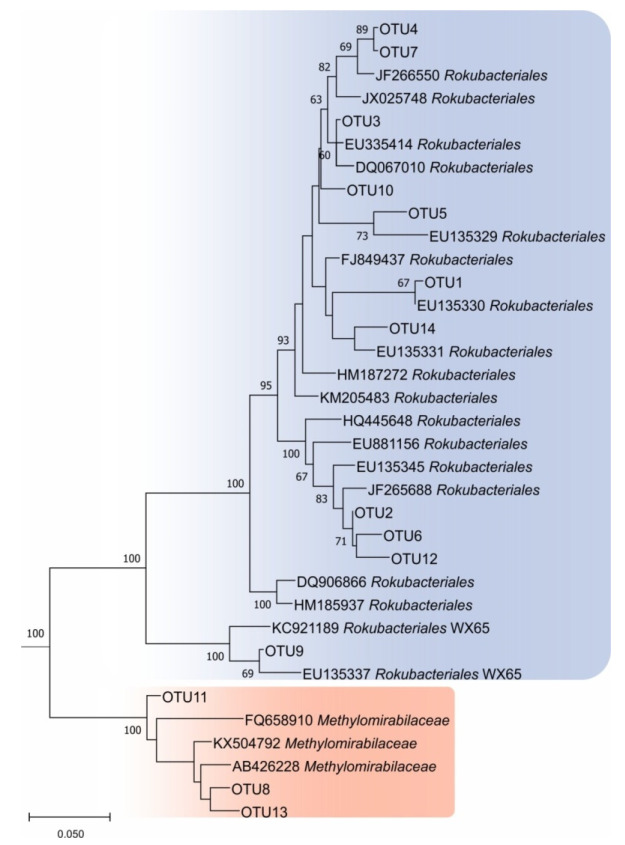
16S rRNA gene-based maximum-likelihood tree showing the phylogenetic relationship of OTUs from Table 2 to representative members of the phylum *Methylomirabilota*. The root (not shown) is composed of five 16S rRNA gene sequences from anammox planctomycetes (AF375994, AF375995, AY254883, AY257181, AY254882). Bar, 0.05 substitutions per nucleotide position.

**Figure 4 microorganisms-10-00011-f004:**
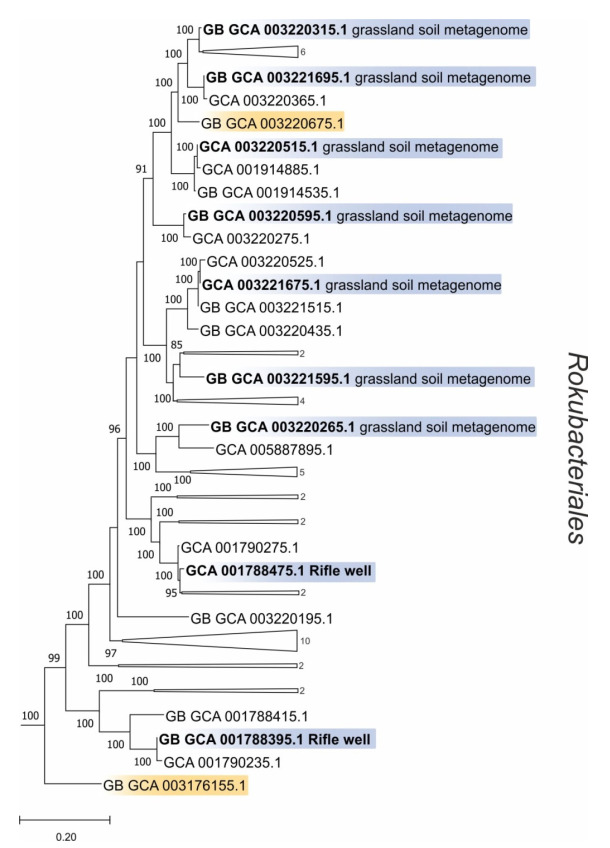
Phylogenomic tree showing the positions of representatives of the *Rokubacteriales* based on the comparative sequence analysis of 120 ubiquitous single-copy proteins. MAGs affiliated with OTUs from this study and MAGs containing CuMMO genes are highlighted by blue and orange, respectively. The significance levels of interior branch points obtained in maximum-likelihood analysis were determined by bootstrap analysis (100 data re-samplings). Bootstrap values of >70% are shown by black circles. The root (not shown) is composed of 210 genomes affiliated with *Omnitrophota* phylum. Scale bar indicates the number of substitutions per amino acid position.

**Figure 5 microorganisms-10-00011-f005:**
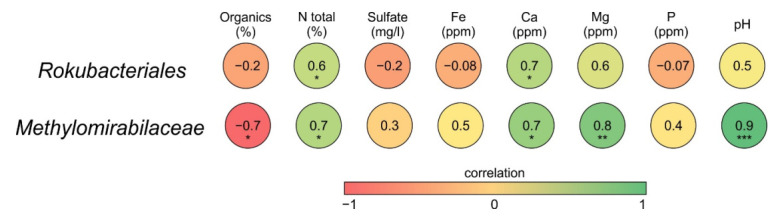
The correlation matrix based on Pearson’s correlation analysis between peat properties and the relative abundances of the *Rokubacteriales* and *Methylomirabilaceae*. The color of circles represents the correlation strength. Significant correlations are indicated by asterisks; *p*-value confidence level * <0.05; ** <0.01; *** <0.001.

**Figure 6 microorganisms-10-00011-f006:**
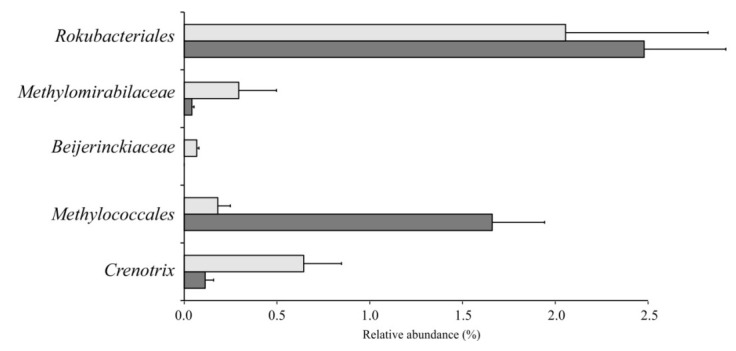
Relative abundances of the *Methylomirabilota* and conventional methanotrophs in peat samples from the fen Shichengskoe before (white bars) and after (grey bars) incubation with 10% methane for 4 weeks, as assessed by Illumina 16S rRNA gene sequencing.

**Figure 7 microorganisms-10-00011-f007:**
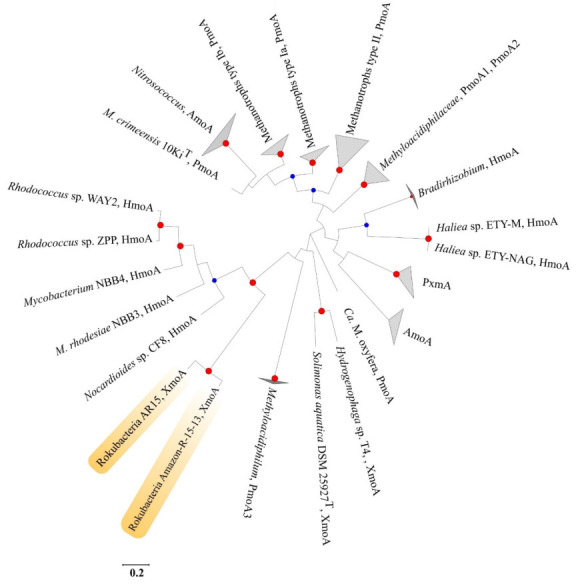
Maximum-likelihood tree showing the phylogenetic relationships between rokubacterial XmoA and XmoA from other members of CuMMO superfamily: ammonia monooxygenases (AmoA), methane monooxygenases (PmoA), hydrocarbon monooxygenases (HmoA). The significance levels of interior branch points were determined by bootstrap analysis (1000 data re-samplings). Bootstrap values are indicated by blue (70–90%) and red (>90%) circles. Bar, 0.2 substitutions per amino acid position.

**Table 1 microorganisms-10-00011-t001:** Characteristics of the peatlands examined in this study.

Mires		Characteristics								
		Coordinates	pH	TOC (%)	N Total (%)	Sulfate (mg/L)	Fe (ppm)	Ca (ppm)	Mg (ppm)	P (ppm)
RAISED BOGS *	1	59°56′56″ N 41°16′59″ E	4.3	88.5	0.605	172	343	3522	634	614
	2	60°46′29″ N 36°49′35″ E	3.7	85.1	0.923	220	1347	4190	682	791
	3	59°27′10″ N 40°30′45″ E	4.3	88	0.685	211	662	4191	905	721
	4	59°22′33″ N 39°59′26″ E	4.1	81.5	1.16	200	5306	3765	816	1020
FENS **	1	59°56′31″ N 41°15′53″ E	7.4	73.6	2.31	202	9387	29,834	2575	1179
	2	60°46′08″ N 36°49′30″ E	6.9	71.6	1.65	222	16,344	27,373	1078	1305
	3	59°47′08″ N 37°52′08″ E	7.6	41.8	1.06	186	106,966	32,196	1599	8920
	4	61°08′18″ N 36°33′27″ E	6.9	83.2	2.55	230	3455	15,968	2583	1049
	5	61°07′16″ N 36°33′21″ E	6.5	48.6	1.51	607	19,264	8494	2665	1192
	6	60°30′42″ N 38°38′59″ E	7.1	66.2	2.4	188	5333	31,193	2695	985

* Raised bogs: (1) Shichengskoe, (2) Piyavochnoe, (3) Alekseevskoe, (4) Barskoe. ** Fens: Shichengskoe (1), Piyavochnoe (2), Radionskoe (3), Ileksa (4), Povreka (5), Charozerskoe (6).

**Table 2 microorganisms-10-00011-t002:** The most abundant OTUs of *Methylomirabilota*-affiliated 16S rRNA gene fragments and their relative abundance in studied fens: 1-Shichengskoe, 2-Piyavochnoe, 3-Rodionskoe, 4-Ileksa, 5-Povreka, 6-Charozerskoe.

OTU	FENS (Relative Abundance %)						Taxon	Close Match	Habitat	Similarity (%)
	1	2	3	4	5	6				
1	1.31	0.00	0.12	0.00	0.00	1.50	*Rokubacteriales*	EU135330	soil grass	97.0
2	0.54	0.00	0.02	0.00	0.00	0.22	*Rokubacteriales*	JF265688	mat lava tube	93.5
3	0.35	0.26	0.03	0.00	0.00	1.01	*Rokubacteriales*	DQ067010	sediment lake	98.8
4	0.24	0.00	0.00	0.00	0.00	0.22	*Rokubacteriales*	JX025748	polluted soil	99.8
5	0.22	0.00	0.00	0.00	0.00	0.00	*Rokubacteriales*	EU135329	soil grass	96.5
6	0.17	0.00	0.00	0.00	0.00	0.07	*Rokubacteriales*	EU135345	soil grass	92.8
7	0.14	0.00	0.13	0.00	0.00	0.31	*Rokubacteriales*	JF266550	mat lava tube	99.3
8	0.12	0.04	0.25	0.25	0.23	0.17	*Methylomirabilaceae*	FQ658910	polluted soil	96.8
9	0.09	0.02	0.00	0.00	0.00	0.03	*Rokubacteriales*_WX65	EU135337	soil grass	86.8
10	0.07	0.00	0.00	0.00	0.00	0.00	*Rokubacteriales*	KM205483	rice rhizosphere	97.5
11	0.00	0.11	0.02	0.00	0.02	0.00	*Methylomirabilaceae*	AB426228	anaerobic field soil	95.5
12	0.00	0.00	0.01	0.00	0.00	0.12	*Rokubacteriales*	JF265688	mat lava tube	92.0
13	0.02	0.00	0.00	0.00	0.00	0.11	*Methylomirabilaceae*	KX504792	sediment lake	96.8
14	0.00	0.00	0.00	0.00	0.00	0.07	*Rokubacteriales*	EU135331	soil grass	96.0

## Data Availability

The raw data generated from sequencing of the 16S rRNA gene fragments before and after methane incubation studies have been deposited in the NCBI Sequence Read Archive (SRA) under the accession numbers SAMN23339836 and SAMN23339837, respectively (BioProject PRJNA782125).

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
