# Peer review of "Rokubacteria in Northern Peatlands: Habitat Preferences and Diversity Patterns"

_microorganisms, 2021, doi:10.3390/microorganisms10010011_

Round 1
Reviewer 1 Report
The article is interesting and of good quality. The title is representative. The abstract contains all the necessary information. The introduction contains all the information necessary to familiarize yourself with the topic. The subject is basic research in the field of environmental microbiology. Methodology correct, clearly presented. The results and their discussion are presented very well and legibly. Please introduce a separate chapter - summary or point-based conclusions.
Author Response
Comment: The article is interesting and of good quality. The title is representative. The abstract contains all the necessary information. The introduction contains all the information necessary to familiarize yourself with the topic. The subject is basic research in the field of environmental microbiology. Methodology correct, clearly presented. The results and their discussion are presented very well and legibly. Please introduce a separate chapter - summary or point-based conclusions.
Replay: We thank the referee for the positive response. The corresponding text section (Conclusions) is included now.
Reviewer 2 Report
Manuscript ID: microorganisms-1508579 by Ivanova et al., entitled " Rokubacteria in northern peatlands: habitat preferences and diversity patterns " aims to assess the abundance of as-yet-uncultivated Rokubacteria in various peatlands, their correlation with environmental conditions and possible links to methanotrophy.
General comments:
The proposed manuscript is well written and structured. The methodology is appropriate and the results are important for characterizing rare microorganisms.
My main comment concerns methane incubation experiments. It is written: "Weighed portions of peat (10 g) ... were placed in glass flasks with a capacity of 160 ml ...". It is then reported that the methane oxidation was calculated in mg CH4 g-1 of wet peat h-1. Was there just a solid sample or a constant humidity or liquid (water) sample? I wonder if the sample has not been dried by cyclic aeration and methane injection. A decrease in humidity can be a very important factor in limiting microbial metabolism. It may also be related to the availability of ingredients, i.e. the content of total nitrogen (TN), Ca and Mg, which turned out to be of some importance for Rokubacteriales. Have you re-injected methane to 10% (v / v) every week? I am interested in what was the methane content at the end of each week of treatment (before re-injection)? This could be an interesting visualization of methanotrophy (week after week) - presented in the form of a figure. I suspect that the natural temperature of peat in June is 15ËšC, and in the remaining months, it is rather around 5-10ËšC. Therefore, I am curious why the experiment used room temperature since it is rather not achieved in real conditions.
Finally, it would be much more reliable for the conclusions if the second sample from Charozerskoe (with possibly a different Rokubacteria population) were also used. Nevertheless, the presented manuscript has good originality, provides knowledge, but there is room for improvement.
Probably an editorial matter, but consider resizing all figures. 2/3 of its current size should be sufficient and easy to read.
Specific comments:
Line 64 add references
Line 73 co-authors
Line 97-100 remove from the introductory part. These sentences are the results/conclusions, given also in the summary paragraph
Line 363 Table S1 is not available from the given link
Lines 194-196 consider removing. Similar information, but detailed is given in lines 217-220
Line 269-275 consider changing to "Relative Rokubacteriales abundance correlated positively with pH, ​​total nitrogen (TN), Ca and Mg availability, and negatively correlated with total organic carbon (TOC), ), sulfate, Fe and P availability (Figure 5). Notably, a strong positive correlation with pH, TN, Mg and Ca availability, as well as a negative correlation with TOC, was observed for the representatives of Methylomirabilaceae."
Author Response
Comment: My main comment concerns methane incubation experiments. It is written: "Weighed portions of peat (10 g) ... were placed in glass flasks with a capacity of 160 ml ...". It is then reported that the methane oxidation was calculated in mg CH4 g-1 of wet peat h-1. Was there just a solid sample or a constant humidity or liquid (water) sample? I wonder if the sample has not been dried by cyclic aeration and methane injection. A decrease in humidity can be a very important factor in limiting microbial metabolism. It may also be related to the availability of ingredients, i.e. the content of total nitrogen (TN), Ca and Mg, which turned out to be of some importance for Rokubacteriales. Have you re-injected methane to 10% (v / v) every week? I am interested in what was the methane content at the end of each week of treatment (before re-injection)? This could be an interesting visualization of methanotrophy (week after week) - presented in the form of a figure. I suspect that the natural temperature of peat in June is 15ËšC, and in the remaining months, it is rather around 5-10ËšC. Therefore, I am curious why the experiment used room temperature since it is rather not achieved in real conditions.
Replay: We have added some details of the procedure in order to clarify this issue. For this experiment, we used native wet peat with high water content (91%). The main purpose of flushing was to remove accumulated CO2 and to keep high methane availability during the whole incubation period. The flushing was very short (one minute only), so that drying of the samples can be excluded. Tracing a decline in CH4 concentration in incubation flasks would have been possible only if we’d use much lower methane concentrations between 0.1 and 1%. No decline could be measured with methane concentration of 10%. Therefore, we’ve made standard activity measurements (with 0.1% methane) before and after one-month incubation period. Regarding the incubation temperature. As shown in our previous studies, 15-20C is the optimal temperature range for methane oxidizing activity in peat (Dedysh, Panikov. Effect of pH, temperature, and concentration of salts on methane oxidation kinetics in Sphagnum peat. Microbiology, 1997, 66(4), 476-479). There was nothing wrong in measuring the activity at 20C.
Comment: Finally, it would be much more reliable for the conclusions if the second sample from Charozerskoe (with possibly a different Rokubacteria population) were also used.
Replay: You’re right but our next chance to obtain fresh peat samples from this alternative location will be in June 2022. We cannot add these data to the manuscript, sorry.
Comment: Probably an editorial matter, but consider resizing all figures. 2/3 of its current size should be sufficient and easy to read.
Replay: Ok, we will take this comment into account while formatting the final manuscript version.
Specific comments:
Comment: Line 64 add references
Replay: The references are added now.
Comment: Line 73 co-authors
Replay: Corrected.
Comment: Line 97-100 remove from the introductory part. These sentences are the results/conclusions, given also in the summary paragraph
Replay: Removed.
Comment: Line 363 Table S1 is not available from the given link
Replay: The table is given in the main text as Table 1 now.
Comment: Lines 194-196 consider removing. Similar information, but detailed is given in lines 217-220
Replay: Ok, suggested lines are removed now.
Comment: Line 269-275 consider changing to "Relative Rokubacteriales abundance correlated positively with pH, ​​total nitrogen (TN), Ca and Mg availability, and negatively correlated with total organic carbon (TOC), ), sulfate, Fe and P availability (Figure 5). Notably, a strong positive correlation with pH, TN, Mg and Ca availability, as well as a negative correlation with TOC, was observed for the representatives of Methylomirabilaceae."
Replay: The corresponding text has been modified as suggested by the reviewer.
Reviewer 3 Report
BRIEF SUMMARY
I present my corrections/remarks below and if authors improve/answer, I could give the “green light” for publication of this work in “Microorganisms” Journal.
SPECIFIC COMMENTS
- Line 2: Begin the second sentence of the title with capital letter, i.e., “Habitat preferences and…”.
- Line 6: Delete “1” before the name of the Institute, i.e., “1 Winogradsky Institute of…”
- Lines 11-12: It is not necessary to repeat all the details of the corresponding author, you can just write: “* Correspondence: dedysh@mail.ru”
- Lines 18-20: Avoid in the whole text the words like “we”, as it sounds selfish. Replace “we examined” with “were examined” and write this after the word “raised bogs”, i.e. “A dataset of 165 rRNA gene reads retrieved from four boreal raised dogs were examined and….”.
- Lines 22-23: Similarly, replace “we incubated” with “were incubated” and write this in the end of the sentence.
- 6. Line 36: Replace “Lipson and Schmidt (2004)” with “[4]”. If you want, you can write the names like this: “Lipson and Schmidt [4]”
- Line 48: Each reference has a number according with its appearance in the text. You write here “Becraft et al. (2017)”, but this is No. 17 in the references. Please change the numbers, this one (Becraft et al.) must be No. 5.
- Lines 94-97: Similarly with previous comments, replace “We also examined” with “were also examined” and write this in the end of the sentence.
- Lines 97-99: Replace “We did not detect”, by following the way of the previous comments.
- Line 112: Where is Table S1?
- Line 266: The title of each paragraph and its text must be in the same page. Also, transfer Line 266 in next page 10.
- Between Lines 363-363: In every work there must be a paragraph with Conclusions, I am surprised that you didn’t write one. Thus, please add “4. Conclusions” and describe briefly your work. Especially you should point the original parts of your work and your contribution to existing science.
I believe that if the authors follow my suggestions, and
answer ALL my questions,
this paper could be suitable for “Microorganisms” Journal.
I would like to check it one more time before the final publication.
Author Response
Comment: Line 2: Begin the second sentence of the title with capital letter, i.e., “Habitat preferences and…”.
Replay: Done. The title has been formatted according to the journal rules.
Comment: Line 6: Delete “1” before the name of the Institute, i.e., “1 Winogradsky Institute of…”
Replay: Done.
Comment: Lines 11-12: It is not necessary to repeat all the details of the corresponding author, you can just write: “* Correspondence: dedysh@mail.ru”
Replay: Corrected.
Comment: Lines 18-20: Avoid in the whole text the words like “we”, as it sounds selfish. Replace “we examined” with “were examined” and write this after the word “raised bogs”, i.e. “A dataset of 165 rRNA gene reads retrieved from four boreal raised dogs were examined and….”. Lines 22-23: Similarly, replace “we incubated” with “were incubated” and write this in the end of the sentence.
Replay: You’re right. Corrected as recommended.
Comment: Line 36: Replace “Lipson and Schmidt (2004)” with “[4]”. If you want, you can write the names like this: “Lipson and Schmidt [4]”
Replay: We have chosen the second option.
Comment: Line 48: Each reference has a number according with its appearance in the text. You write here “Becraft et al. (2017)”, but this is No. 17 in the references. Please change the numbers, this one (Becraft et al.) must be No. 5.
Replay: Becraft et al is now changed to No. 5 and the corresponding reference is included in the list.
Comment: Lines 94-97: Similarly with previous comments, replace “We also examined” with “were also examined” and write this in the end of the sentence.
Replay: Done.
Comment: Lines 97-99: Replace “We did not detect”, by following the way of the previous comments.
Replay: This part is now moved to Conclusions in detached impersonal style as was asked by second reviewer.
Comment: Line 112: Where is Table S1?
Replay: This table is included in the main text as Table 1 now.
Comment: Line 266: The title of each paragraph and its text must be in the same page. Also, transfer Line 266 in next page 10.
Replay: Thank you, the text is formatted now in a proper way.
Comment: Between Lines 363-363: In every work there must be a paragraph with Conclusions, I am surprised that you didn’t write one. Thus, please add “4. Conclusions” and describe briefly your work. Especially you should point the original parts of your work and your contribution to existing science.
Replay: This was also requested by Reviewer 1. Done.